# Consistency Is Key When Setting a New World Record for Running 10 Marathons in 10 Days

**DOI:** 10.3390/ijerph182212066

**Published:** 2021-11-17

**Authors:** Nicolas Berger, Daniel Cooley, Michael Graham, Claire Harrison, Georgia Campbell, Russ Best

**Affiliations:** 1School of Health and Life Sciences, Teesside University, Borough Road, Middlesbrough TS1 3BA, UK; D.Cooley@tees.ac.uk (D.C.); Michael.Graham@tees.ac.uk (M.G.); G.Campbell@tees.ac.uk (G.C.); 2Newcastle Nutrition Community Team, The Newcastle upon Tyne Hospitals NHS Foundation Trust, Newcastle NE7 7AH, UK; claire@c-harrison.co.uk; 3Centre for Sports Science & Human Performance, WINTEC, Hamilton 3240, New Zealand; Russell.Best@wintec.ac.nz

**Keywords:** ultra-endurance, Marathon, pacing, energy expenditure, world record, running

## Abstract

Background: We describe the requirements and physiological changes when running 10 consecutive marathons in 10 days at the same consistent pace by a female ultra-endurance athlete. Methods: Sharon Gayter (SG) 54 yrs, 162.5 cm, 49.3 kg maximal oxygen uptake (VO_2_ max) 53 mL/kg^−1^/min^−1^. SG completed 42.195 km on a treadmill every day for 10 days. We measured heart rate (HR), Rating of Perceived Exertion (RPE), oxygen uptake (VO_2_), weight, body composition, blood parameters, nutrition, and hydration. Results: SG broke the previous record by ~2.5 h, with a cumulative completion time of 43 h 51 min 39 s. Over the 10 days, weight decreased from 51 kg to 48.4 kg, bodyfat mass from 9.1 kg to 7.2 kg (17.9% to 14.8%), and muscle mass from 23.2 kg to 22.8 kg. For all marathons combined, exercise intensity was ~60% VO_2_ max; VO_2_ 1.6 ± 0.1 L.min^−1^/32.3 ± 1.1 mL.kg^−1^.min^−1^, RER 0.8 ± 0, HR 143 ± 4 b.min^−1^. Energy expenditure (EE) was 2030 ± 82 kcal/marathon, total EE for 10 days (including BMR) was 33,056 kcal, daily energy intake (EI) 2036 ± 418 kcal (20,356 kcal total), resulting an energy deficit (ED) of 12,700 kcal. Discussion: Performance and pacing were highly consistent across all 10 marathons without any substantial physiological decrements. Although overall EI did not match EE, leading to a significant ED, resulting in a 2.6 kg weight loss and decreases in bodyfat and skeletal muscle mass, this did not affect performance.

## 1. Introduction

Historically the most famous ultra-endurance feat was that of the dispatch runner Pheidippides, who ran from Marathon to Athens to deliver the message of the victory of the battle of Marathon in 530 BC. He died immediately after delivering the message of “Joy to you, we’ve won!”, which seems surprising as the distance from Athens to Marathon is ‘just’ 40 km. However, Pheidippides had previously already covered 480 km in two days when running to and from Sparta to request help when the Persians landed at Marathon [1]. Thus, although the rather more modest distance of 40 km underplays his stamina, the fact that Pheidippides died whilst carrying out his duties has defined this now-famous distance.

Pheidippides was a *hemerodromos*, who were day-long runners in the Greek military who covered large distances on foot, often over difficult mountainous terrain. It has been reported that they often did not sleep for days to be able to carry out their duty and deliver messages [2]. Their nutrition was one high in carbohydrates, protein, fat and salt, which they consumed via figs and other fruit, olives, dried meats and a very early version of the energy gel called pasteli. This was a paste made from ground sesame seeds and honey. The day-long runners also ate small amounts of sea buckthorn, which they believed enhanced endurance and stamina [1].

Today, ultra-running (UR) is no longer required to deliver messages, but people typically take part because of the extremely challenging nature. UR is considered any running event in excess of a marathon (>42.2 km; IAAF, 1997). Races are contested over standardised race distances (50 km, 50 miles, 100 km and 100 miles (IAU, 2007)), or time periods (e.g., 24 h) in either single-day, or multi-day events. UR events may take place on a variety of terrains, ranging from treadmills to remote wilderness, and has increased in popularity and increasingly, more research about the impact of these events on the human body is being carried out [3]. However, there is still only limited literature, which includes data from successful world record attempts, or that includes all necessary data to calculate energy expenditure (EE), such as VO_2_, or pre and post-haematological and body composition data. Typically, they include only one of the aforementioned measures or estimates, e.g., of energy expenditure [4,5]. Therefore, we set out to collect detailed data on a female multiple world record-holding ultra-runner during her successful attempt at breaking the world record for running 10 marathons in 10 days on a treadmill.

It is still not fully clear what effects repeated ultra-endurance performances have on the body, especially in relation to changes in body mass and corresponding body composition, in particular, muscle mass and fat-free mass [3]. Matching EE during UR is often problematic due to logistical constraints and the extremely common gastro-intestinal complaints suffered by competitors [6,7,8]. Physiological problems arising after UR nearly always include decreases in body mass and dehydration, loss of skeletal muscle mass and increases in total body water [9,10]. This trend is also apparent in the back-to-back triathlon format of Ultraman, which consists of completing multiple long-distance triathlons over consecutive days [11,12] UR has also been shown to result in haemolysis [13] and haematocrit can decrease [14,15], although a reduction in haemoglobin is typically the result of plasma volume expansion [15].

Pacing and consistent effort in UR are key to performance and it has been shown that the most successful UR athletes set and follow a pre-determined pacing strategy, and inexperienced and/or younger runners tend to start faster and fade as the event progresses [16]. Being able to perform repeatedly and consistently over several days requires careful planning and well-organized recovery strategies. With inadequate recovery, there will be significant decrements in bodily systems, which could affect performance [17]. Recovery strategies include carbohydrate and protein intake to replenish depleted glycogen stores and enhance repair from muscle damage, as well as manual therapies, such as massage or ice baths [18,19].

The aim of the present study was to investigate the effects of running 10 marathons (total 421.95 km) on 10 consecutive days on changes in body composition and haematology, to examine the pacing strategy, calculate the energy intake and expenditure, and analyse recovery methods. There were follow-up measurements 1-day and 1-week post-completion of the final marathon. We expected decreases in body fat and muscle mass, as well as a significant energy deficit, but no haematological changes.

## 2. Materials and Methods

*Participant*: Sharon Gayter (SG) is a 57-year-old ultra-endurance runner (54 yrs at the time of the event) with over 30 years’ competitive experience. She has competed internationally for GB at 100 km and 24 h, completed more than 50,000 km of racing, finished more than 1700 races, and finished more than 400 marathons and over 160 ultra-endurance events.

### 2.1. Record Attempt

SG successfully completed a world record attempt in accordance with Guinness World Record requirements. To be eligible for a record, efforts must be based upon one variable, breakable, measurable, standardisable, and verifiable (Guinness World Records Ltd., 2019) with supporting evidence submitted as per Guinness World Records requirements (Guinness World Records Ltd., 2019). The record involved the completion of, and cumulative time for 10 marathons in 10 days completed on a treadmill in the UK at Teesside University.

SG started running every day at 11 am on a treadmill located in the foyer of Teesside University and did not stop running until she completed the full distance of 42.195 km. Temperature was kept at a constant 20 °C throughout the attempt. We recorded weight (kg), body composition, sleep data, lung function, blood lactate (mmol.L^−1^), blood glucose (mmol.L^−1^), cholesterol, haemoglobin (g.dl) and haematocrit (%) before and after each marathon, 1-day and 1-week post-attempt.

### 2.2. Measures

#### 2.2.1. Body Composition

Body mass (Seca 869, Birmingham, UK) and body composition were assessed via a portable bioelectrical impedance analysis (BIA) machine (InBody S10, 13,850 Cerritos Corp Dr, Unit C, Cerritos, CA, USA) to determine body mass (BM), skeletal muscle mass (SM), percent body fat (%BF) and total percent body water (%TBW). Measurements were taken before the start of the attempt, at the same time of day (10.30 a.m.) before and immediately after each marathon, as well as 1-day and 1-week post-attempt completion. BIA measurements were performed with the athlete standing in an upright position, barefoot, with legs and thighs not touching, and the arms not touching the torso. A total of 6 electrodes in total were attached to: the thumb and middle finger of both hands and the ankles of both feet. The skin and the electrodes were pre-cleaned and dried prior to use. Reliability of the InBody S10 has previously been shown to be high within and between users but may underestimate body fat mass [19]. With Inbody S10 basal metabolic rate (BMR) was estimated by using the regression equation BMR = 370 + 21.6 × FFM (fat-free mass).

#### 2.2.2. Pre-Attempt Laboratory Testing

Prior to the world record attempt, SG completed a VO_2_ max test to assess her maximum oxygen uptake and lactate threshold. The test was performed on the same treadmill as used for the world record attempt (Quasar, H/P/Cosmos, H/P/Cosmos Sports and Medical GMBH, Germany). The athlete started running at 11 km/h and a gradient of 1%. The speed was increased by 1 km/h every 3 min until the speed could no longer be maintained, i.e., volitional exhaustion [20]. Breath-by-breath measures of VO_2_, and carbon dioxide output (VCO_2_) were measured continuously (Piston HD6000, nSpire, nSpire Health Inc, Hertford, UK). The gas analyser was calibrated according to the manufacturer’s instructions with known gases, and the volume transducer was calibrated with a 3-litre calibration syringe (Hans Rudolph 5530, Shawnee, OH, USA). Heart rate was measured continuously by telemetry and noted down in the last minute of every stage (PE4000, Polar Electro Oy, Kempele, Finland) when a steady-state HR was reached. At the end of every stage the athlete straddled the treadmill, and a fingertip blood sample was taken for immediate analysis of blood lactate (Ysi, 2300 Lactate and Glucose, Ysi UK Ltd., Hampshire, UK).

#### 2.2.3. Cardio-Respiratory, Heart Rate and Subjective Measures

A heart rate monitor was worn continuously (PE4000, Polar Electro Oy, Kempele, Finland) and heart rate was measured once in the final minutes each hour. VO_2_ was collected for 2 min in the final stages of each marathon using a breath by breath online portable gas analyser (MetaMax 3b, Cortex GmbH, Leipzig, Germany). The record was completed on a standard treadmill (Quasar, H/P/Cosmos, H/P/Cosmos Sports and Medical GMBH, Germany) situated in the open area of the Olympia building of Teesside University in the UK. The average of each 2-min sample was calculated from the raw, unfiltered, breath by breath data. From measures of VCO_2_ and VO_2_, total carbohydrate (CHO) and fat oxidation rates were calculated using the formulas of Frayn [21], assuming negligible protein oxidation:Total fat oxidation (g/min^−1^) = 1.67 ∗ VO_2_ − 1.67 ∗ VCO_2_
Total CHO oxidation (g/min^−1^) = 4.55 ∗ VCO_2_ − 3.21 ∗ VO_2_

Rating of perceived exertion (RPE) was assessed via the CR-10 scale [22]. RPE was recorded every hour. Arbitrary units are reported with accompanying verbal descriptors.

#### 2.2.4. Sleep and Lung Function

Resting HR and sleep metrics were tracked throughout via Garmin Vivoactive (Garmin International, Olathe, KS, USA). As SG is asthmatic, peak flow rate (PFR) and peak expiratory flow (PEF) were measured every morning upon waking (Vitalograph, Lenexa, KS, USA).

#### 2.2.5. Haematology and Body Composition

All blood samples were taken pre and post every marathon from a fingertip capillary from the ring finger of the left hand, prior to analysis. Blood lactate (mmol.L^−1^), (Ysi, 2300 Lactate and Glucose, Ysi UK Ltd., Hampshire, UK), haemoglobin (g.dl) (Prospect Hb, Diaspect Medial GMBH, Von-cancrin-str1, 63,877 Salauf, Germany), glucose (mmol.L^−1^) (NESCO multicheck, Kernel int’l Corp, Taiwan) and cholesterol levels (HDL, LDL, Trig; Mission Cholesterol Meter, Acon Labs Inc, 10,125 Mesa Rim Road, San Diego, CA, USA).

Samples for haematocrit (%Hct) were collected in heparinised glass tubes and spun for 5 min at 14,000 rpm (Haemotrocit Centrifuge, Sigma, 1-14, Sigma Laboratories, D-37520 Osterode am Harz, Germany). Spun blood was analysed using a Hawksley micro haematocrit reader (Hawksley, Lancing, Sussex, UK).

#### 2.2.6. Nutritional Intake

Nutritional intake was recorded in a detailed diary during the attempt, and the intake was analysed for total energy (kcal) and intakes for macronutrients per day using specialist software (Nutritics v.x, Nutritics, Dublin, Ireland). Relative intake (g·kg^−1^) was also calculated for comparison to sports nutrition guidelines and previous research.

### 2.3. Other Recovery Measures

Following the post-run blood and body composition measures the athlete had an ice bath for her feet and a massage every day.

### 2.4. Statistical Analysis

Data are reported as both means ± standard deviations and per marathon (Figure 1, Figure 2, Figure 3 and Figure 4; Table 1 and Table 2). Data were stored, compiled and analysed in Microsoft Excel (v16.39; 2020; Microsoft, Redmond, Washington, DC, USA) due to the relatively simplistic approach to statistical testing employed.

## 3. Results

### 3.1. Laboratory Testing before the Attempt

In November 2017, SG was 162.5 cm, weighed 49.3 kg and reached a top running speed of 16 km/h. Her was VO_2_ max 53 mL/kg^−1^/min^−1^ (2.59 L/min^−1^), with her lactate threshold occurring at 75% of VO_2_ max. Her maximal heart rate was 174 b.min^−1^.

### 3.2. Record Breaking Performance

SG set out to beat the previous record and targeted sub 45 h. SG completed 10 marathons in 10 consecutive days in a total time of 43 h 51 min 39 s, breaking the previous world record of 46 h 15 min by 2 h 24 min, running 2.5 h faster than the previous record. She ran consistently at 9.2–9.7 km/h and varied her speed by 0.1–0.3 km/h every ~10 min (see Table 1 for individual marathon times).

### 3.3. Body Composition

Pre attempt to post attempt SG’s weight decreased from 51 kg to 48.4 kg, bodyfat mass decreased from 9.1 kg to 7.2 kg, bodyfat percent decreased from 17.9% to 14.8%, muscle mass decreased from 23.2 kg to 22.8 kg. Body mass decreased by 2.6 kg at the end of the event (−5%), with an average daily loss of 0.9 ± 0.3 kg, or 1.8% total body weight. Skeletal muscle mass decreased by 0.4 kg at the end of the event (−1.72%), body fat mass decreased by 1.9 kg (−20.88%), percent body fat decreased by 3.1% (−17.32%), and visceral fat decreased by 8 cm^2^ (−25.81%). Intracellular and extracellular water showed little change and decreased by 0.3% (−1.55% total) and 0.2% (−1.77% total), respectively.

### 3.4. Heart Rate

Mean HR for all marathons combined was 143 ± 4 b.min^−1^. For all 10 marathons combined heart rate was similar between the 1st and 2nd hours before increasing in the 3rd and 4th hours for each marathon, on average these increases were~4–5% in both instances (see Table 2 and Table 3 for detailed heart rate data).

### 3.5. Sleep and Resting Heart Rate

SG’s resting HR increased slightly from 51 b.min^−1^ on the first day to 53–57 b.min^−1^ on the remaining days. Her average resting HR was 54 ± 1.6 b.min^−1^. Mean sleep time was 500.6 ± 55.7 min (8 h 20 min), with the longest sleep time 9 h 47 min on day 7, and the shortest 6 h 50 min on day 4. The time spent in deep sleep fluctuated from a peak 3 h 33 min on day 10, and the shortest was 1 h 0 min on day 8.

### 3.6. Lung Function, Oxygen Uptake and Energy Expenditure

Peak flow rate (PFR)/peak expiratory flow (PEF) decreased slightly, with a mean PFR/PEF of 379 ± 18.1 L/min^−1^, and the highest PFR of 410 L/min^−1^ on day 1 and the lowest of 350 L/min^−1^ on days 2 and day 6.

VO_2_ was similar for all 10 marathons at 1.6 ± 0.1 L.min^−1^/32.3 ± 1.1 mL.kg^−1^.min^−1^. With a VO_2_ max of 53 mL/kg^−1^/min^−1^/2.588 L/min^−1^, SG worked at ~60% of her VO_2_ max during every marathon.

Basal metabolic rate (BMR) was calculated as 1274 kcal via BIA. Energy expenditure (EE) averaged 2030 ± 82 kcal/marathon and thus totalled of 20,306 kcal for all 10 marathons. Her average RER was 0.8 ± 0 and, therefore, 33.4% of calories came from carbohydrate (CHO), 66.6% came from fat oxidation. She utilised 157 ± 6 g of CHO and 147 ± 6 g of fat during each marathon and metabolised a total of 1570 g of CHO and 1468 g of fat.

Mean daily energy intake (EI) was 2036 ± 418 kcal, with total 20,356 kcal over the 7 days. EI almost exactly matched EE from the marathons but did not match total daily EE including basal metabolic rate of 1274 kcal. Total overall EE including marathons and BMR was 33,056 kcal, leading to an energy deficit (ED) of −12,700 kcal.

### 3.7. Nutritional Intake

Energy and absolute and relative macronutrient intakes for each day of the 10 days are presented in Figure 1 and Figure 2; Table 4 provides an example of the nutritional intake for a single day.

Calorie consumption during each run decreased significantly after the first two days due to complaints of nausea. Nausea was treated with motion sickness medication (day 1 and 2; Cinnarizine) and pressure point wrist bands for the remaining days. On days 1 and 2, SG consumed a carbohydrate solution with a small amount of added protein (day 1: 180 kcal; day 2: 126 kcal). The following 8 days she only consumed plain water and 11–25 grapes during each run, with a mean of 46.4 ± 25 kcal/marathon, a total of 371 kcal from grapes. For all 10 days the calories consumed during the runs were 67.7 ± 51.5 kcal/day: a total intake of 676.5 kcal.

The mean daily total calorie consumption over the 10 days was 2035.6 ± 418.4 kcal, with the lowest overall intake on day 1 with 1468 kcal and the highest on day 6 with 2688 kcal. EE was 2030 ± 82 kcal/marathon, total EE for 10 days (including BMR) was 33,056 kcal, daily EI 2036 ± 418 kcal (20,356 kcal total), resulting an ED of 12,700 kcal (Figure 1). Average intake was 279.9 g.day^−1^ for CHO (5.7 g.kg^−1^), 112.9 g day^−1^ protein (2.3 g.kg^−1^), and 51.8 g day^−1^ fat (1.1 g.kg^−1^) (Figure 2 and Figure 3). Overall, 55% of calories consumed came from CHO, 22.2% came from protein and 22.9% came from fat (Figure 4).

Fluid intake was consistent over the 10 days at 4134 ± 313 mL/day, divided into 2309 ± 280 mL/day when not running and 1825 ± 206 mL/day when running. While running, in the first 3 days fluid intakes were 2250 mL, 2000 mL and 2000 mL, after which it reduced to 1750 mL every day, apart from day 8 when it reduced to 1500 mL. The decrease was due to the cessation of CHO solution consumption. In total, SG consumed 41,340 mL over the 10 days: 23,090 mL when not running and 18,250 mL when running.

### 3.8. Haematology

Haemoglobin (Hb) was 13.2 g.dl and haematocrit (Hct) were 39% on day 1. On average, Hb increased from 13.2 ± 0.5 g.dl to 13.7 ± 0.6 g.dl, (3.6% increase), and Hct increased from 38.6 ± 1.0% to 40.2 ± 1.2% (4.2% increase). 24 h post completion Hb was 11.4 g.dl, and Hct 37%, and 7 days post event Hb was 13.9 g.dl and Hct was 42%.

Blood lactate prior to starting each marathon was 1.1 ± 0.3 mmol.L^−1^ and post was 1.3 ± 0.3 mmol.L^−1^, demonstrating a negligible excursion of 0.2 mmol.L^−1^ pre to post for all 10 marathons.

Blood glucose remained within normal range pre to post for each marathon, dropping from 6.1 ± 1.0 mmol.L^−1^ to 5.4 ± 0.4 mmol.L^−1^, a change of 9.9%. 24 h post glucose was 6.0 mmol.L^−1^ and 7 days post it was 4.5 mmol.L^−1^.

LDL cholesterol decreased pre to post for all 10 days from 2.3 ± 0.3 mmol.L^−1^ to 1.8 ± 0.4 mmol.L^−1^ (20.7% decrease). 24 h post LDL was 2.59 mmol.L^−1^ and 7 days post was 3.98 mmol.L^−1^. HDL increased trivially pre to post for all 10 days from 2.1 ± 0.2 mmol.L^−1^ to 2.2 ± 0.2 mmol.L^−1^, (6% increase), and continued to decrease 24 h post (1.97 mmol.L^−1^) and 7 days post (1.47 mmol.L^−1^). Triglycerides increased markedly pre to post for all 10 days from 0.8 ± 0.1 mmol.L^−1^ to 1.9 ± 0.3 mmol.L^−1^ (133.4% increase). 24 h post triglycerides were 1 mmol.L^−1^ and 7 days post they were 1.63 mmol.L^−1^.

### 3.9. Subjective Measures

RPE remained stable throughout the 10 days, with a mean rating of 5.0 ± 0.0 arbitrary units for the first hour, 5.0 ± 0.16 for the second hour, 5.17 ± 0.35 for the third hour and 5.56 ± 0.46 for the fourth hour of every day; the equivalent verbal descriptor of an RPE of 5 is *Hard* [19].

## 4. Discussion

The aim of this study was to analyse the energy intake (EI), energy expenditure (EE) and subsequent energy deficit (ED), and track the changes in body composition and haematology, over the course of 10 marathons (421.95 km) ran on consecutive days. We also set out to describe the ideal strategy necessary to set a successful world record (WR). The previous WR of 46 h 15 min was improved by~2.5 h and SG completed the event in 43 h 51 min 39 s.

The main finding from this study was that SG was in a daily ED of 1270 kcal, with a total ED of 12,700 kcal over the 10 days. As a result, SG lost 0.4 kg of skeletal muscle mass and 1.9 kg bodyfat mass. The fact that she consumed limited nutrition during the marathons exacerbated the ED and the loss of body mass but served to minimise nausea and thus potential gastrointestinal distress. The second finding was that the successful strategy SG adopted was a highly consistent pacing strategy and followed a set recovery regime, which included sleeping for an average of 8 h 20 min each night and emphasising dietary carbohydrate intake between marathons. She completed each marathon in between 4 h 21 min and 4 h 24 min. She ran unfailingly at 9.2–9.7 km/h and varied her speed by 0.1–0.3 km/h at approximately 10 min intervals to ensure a small change in pace and to relieve monotony. Each marathon was followed by a protein and carbohydrate recovery drink and a massage [19].

### 4.1. Nutritional Intake and Energy Deficit

Recommendations for CHO intake in highly active athletes are 5–8 g/kg/day [23,24]. SG fulfilled this recommendation by consuming 5.7 ± 1.0 g/kg/day, although recommendations for those in high volume intense activity of 3–6 h/day may need to consume as much as 8–10 g/kg/day [23,24]. The dietary recommendations for fat for athletes are similar to those for non-athletes in promoting health, constituting ~30% of daily caloric intake [25]. SG only consumed 22.9% (1.1 ± 0.4 g/kg/day) of her total calories from fat, although this is not surprising as she was purposefully emphasising high CHO and protein foods.

Although SG set out to consume CHO throughout the attempt in the form of CHO drinks, smoothies, sweets (jellybeans) and solid food (sandwiches), she complained of GI distress, and significantly altered her nutrition strategy to include almost no food (a small number of grapes) and only plain water after the first two marathons. During the first two marathons, despite the available selection of food and drink, she did not eat any solid food, and only consumed a carbohydrate solution (~150 kcal) in addition to water. Following her nausea, she altered her nutrition strategy again for the following 8 marathons. During each marathon EI was minimal (mean 68 ± 51 kcal/marathon from grapes), instead relying on increasing her EI before and after each run.

It has been suggested that athletes ingest up to 90 g/h of CHO during activities lasting longer than 2.5 h [26,27]. This is especially important when competing back-to-back over several days [28]. Indeed, a recent study by Viribay and colleagues [29] demonstrated that an even higher intake of 120 g/h could limit exercise-induced muscle damage and shorten recovery time, but the gastrointestinal burden of consuming such a high intake likely requires a comprehensive nutritional training strategy [26,30,31] and resultant phenotypic adaptations within the gut. Such high doses have also been reported in cycling [32], but cycling lacks the mechanical load imparted by running, thus facilitating high rates of CHO consumption with lesser risk to inducing GI symptoms, as experienced by SG during the present attempt. SG employed a similar low-CHO-intake strategy during a previously successful world-record attempt [33].

GI complaints are very common in runners, especially those competing over longer distances, with 30–50% being reported in marathon running, 83% at 60 km, and up to 96% in a 161 km ultramarathon [8]. These symptoms may be responsible for race withdrawal and a failure to attain planned nutritional strategies [34]. SG complained of nausea during the 2nd and 3rd day and had to reduce her running speed as she thought she might vomit. Following this she took sea sickness tablets (Sturgeon; Cinnarizine 15 mg) and also wore wrist bands that are alleged to provide acupressure to reduce nausea. From the 5th day onwards, she only wore the wrist bands and no longer took any medication. The nausea seemed to be caused by the carbohydrate drink, but also due to the motion of the treadmill, and as a precaution she only consumed plain water from days 3–10.

Recommendations for protein intake for athletes involved in moderate amounts of exercise are 1.2–2.0 g/kg/day and slightly higher for athletes involved in high volume or intense exercise, who should consume 1.7–2.2 g/kg/day [18,35,36]. In addition, recommendations for older athletes are for single doses of 40 g, especially post-exercise, as older musculature responds more slowly and has lower sensitivity to protein ingestion [37,38,39]. SG’s mean protein intake was 2.3 ± 0.5 g/kg/day, satisfying the recommended daily intake. She achieved this with the addition of protein drinks immediately post-marathon, followed by a large protein bolus at her evening meals. Despite this adequate protein supply, she still lost 1.2 kg in total bodyweight and 0.3 kg of muscle mass. Although we typically assume minimal contribution from protein as an energy source during exercise, these data suggest that protein oxidation took place to the extent that body composition was affected [40,41]. It is unclear whether this increase in protein contribution to metabolism occurred during or between marathons and would have likely been exacerbated by SG’s accumulation of a large ED, but partially offset by the exercise stimulus and EI.

The majority of performance and practices associated with the attempt were the athlete’s choice alone; it is important to note however, that SG is an incredibly experienced ultra-athlete and also engages with sports science literature to support her decision making. SG’s blended use of both tacit and explicit evidence is considered evidence-informed performance. We believe as sports scientists involved in supporting the documented attempt, our role was to assess, analyse, and where appropriate, inform the athlete’s choices, but not to interfere with or direct performance or decision making for the sake of publication.

### 4.2. Body Composition Changes

From the baseline measure before the first marathon to completion of the tenth marathon, SG’s body mass decreased by 2.6 kg (−5%), with an average daily loss of 0.9 ± 0.3 kg (1.8% total body weight). Using BIA, we measured decreases in skeletal muscle mass (0.4 kg; −1.72%), body fat mass (1.9 kg; −20.88%), percent body fat (3.1%; −17.32%), and visceral fat (−8 cm^2^; −25.81% change from pre-event) from the first to the tenth day. We can be relatively confident in these results as the InBody S10 has been shown to be a valid measure of lean body mass and body fat mass in comparison to dual-energy X-ray absorptiometry [39] and is considered highly reliable when used by the same assessor (ICC: 0.89; 95% CI: 0.86–0.92; [42,43]).

In a 5-day, 339 km race Knechtle et al. [4] found no reduction in body mass or fat mass but reported a similar reduction in skeletal muscle mass of 0.63 ± 0.79 kg at the end of the race. This is likely due to the fact that competitors during this event consumed more calories during the event and sustained a much smaller ED than SG.

The results suggest a high contribution from endogenous fat stores towards energy metabolism and a small amount of protein catabolism [44]; this is further supported by SG’s negligible EI during the marathon’s, as fasted training has been shown to increase fat oxidation and branched-chain amino acid oxidation, concomitantly elevating protein catabolism [45,46].

Intracellular and extracellular water remained mostly stable following each marathon and decreased by 0.3% (−1.55% total) and 0.2% (−1.77% total), respectively, indicating that fluid intake (1825 ± 206 mL) during each run matched sweat rate accurately. SG is an experienced runner and had a very consistent fluid intake, equal to 456 mL/h during each marathon. This is ~50% more than reported for ad libitum water consumption during a 100 km race [47], suggesting SG drank to a plan, as opposed to drinking to thirst. Elite runners have been shown to ingest 550 ± 340 mL/h (range 30–1090 mL/h), which is above SG’s consumption, despite a higher running pace throughout [48]. Lower fluid intakes may also be advantageous to avoid hyponatraemia e.g., Kipps et al. [49] reported fluid intakes ranging from 3683 mL/843 mL/h for hyponatraemic runners and 1924 mL/451 mL/h for those without hyponatraemia during the London marathon.

### 4.3. Respiratory Variables

It was reported that UR’s have more allergies (~25.1%) and exercise-induced asthma (∼13.0%) than the general population [50]. SG has pronounced exercise-induced asthma, which she controls with the appropriate medication. She recorded her peak flow rate (PFR) every morning before the start of each marathon, and there was a decrease from 410 L/min^−1^ on day 1 to 350 L/min^−1^ on days 2 and day 6, with an average of 379 ± 18.1 L/min^−1^, although this minor change caused no problems in terms of health or performance.

SG exhibited a consistently low RER of 0.8 ± 0 throughout the attempt. Factors that can influence RER include diet, exercise intensity and duration, muscle glycogen content, proportion of type I muscle fibres, dietary fat intake, training status and blood metabolites (plasma lactate; serum FFA concentrations [44,51]). Fat oxidation increases from ~35% VO_2_ max to a maximal rate at an intensity of 48 ± 1% VO_2_ max [44]. During low-intensity exercise lipids provide more than half of the energy contribution, and as exercise intensity increases, thus does the contribution from CHO, and that from lipids decreases. The reported cross-over point reportedly lies between 48–53% VO_2_ max, or at approx. 50% as suggested previously by others [52].

SG exercised at approx. 60% VO_2_ max with a consistent RER of 0.80, demonstrating a lower RER than previously described in untrained males (0.83–0.95; [53]), confirming a high contribution of fat metabolism towards energy production, despite an increased exercise intensity. There is considerable interindividual variation in substrate utilisation, even at comparatively low work rates e.g., Helge et al. [53] demonstrated that even at 55% VO_2_ max the RER in untrained men varied from 0.83 to 0.95. For trained persons and over a variety of work rates (25, 50 and 70% peak power output) an even larger variability of 0.72–0.93 was shown [51]. Greater variability than observed was expected in SG’s measures given the presumably additive nature of the challenge, but her training status and phenotypic adaptations to years of ultra-running allowed for a predominantly fat derived effort, despite performing at a higher relative intensity than previous experimental work.

### 4.4. Recovery Strategies

Recovery between repeated endurance events is crucial. Strategies include rehydration, refuelling, massage, and sleep [28,54]. Immediately post-marathon (after post-exercise blood samples and body composition were recorded), SG consumed a carbohydrate and protein recovery drink with added colostrum while she had a massage. Massage has been shown to aid recovery, although not to improve performance [17]. SG chose to consume a large evening meal, followed by a dessert and protein and CHO rich milky drink before going to sleep to aid muscle repair, reduce DOMS and increase muscle and liver glycogen replenishment [18].

Sleeping sufficiently is also crucial to recovery [54], and although moderate amounts of exercise enhance sleep quality, high training or racing loads might negatively affect sleep and reduce the ability to recover adequately. Part of the strategy for a successful WR attempt was to maximise sleep and allow enhanced recovery. Sleep time and deep sleep were tracked with a wearable device (Garmin Vivoactive; Garmin, Olathe, KA, USA) and showed that SG slept for 500.6 ± 55.7 min/day (8 h 20 min), with the longest sleep time 9 h 47 min on day 7, and the shortest 6 h 50 min on day 4. The time spent in deep sleep fluctuated from a peak 3 h 33 min on day 10, and the shortest was 1 h 0 min on day 8. Time spent asleep was longer than reported by Knufinke et al. [54] (7 h 50 min) and especially deep sleep was far longer than the reported 1 h 36 min, apart from day 8, and is likely a contributory factor to the stable performance over the 10 days.

### 4.5. Other Parameters

Blood glucose decreased from 6.1 mmol.L^−1^ pre run to 5.4 mmol.L^−1^ post-run; somewhat surprisingly SG was not hypoglycaemic following completion of each marathon despite very low calorie and CHO intake during each run (67.7 ± 51.5 kcal). A more notable decrease would be expected when running for more than 4 h but suggests a sparing of glycogen utilisation in favour of fat oxidation in our athlete [55], and/or maintenance via gluconeogenic pathways.

We observed a 2% decrease in Hct (from 39% to 37%) from the first to the tenth day, but no change in Hb, in contrast to previous findings by Knechtle et al. [47], who found a 1.5 ± 3.5% change in Hct after a 100 km race. This suggests an absence of plasma volume expansion, which is the usual cause for decreases in Hb in long endurance events [15].

For all 10 marathons combined the heart rate was the same for the 1st and 2nd hour of each marathon and then increased for the 3rd and 4th hour for each marathon and averaged 137 ± 6 b.min^−1^ 1st hour, 137 ± 5 b.min^−1^ 2nd hour, 145 ± 6 b.min^−1^ 3rd hour, and 151 ± 5 b.min^−1^ 4th hours. This increase in HR indicates a slight increase in the cost of running and fatigue as there was no increase in pace from beginning to end [56]. The cost of running is measured either as the oxygen required (in mL.kg^−1^.min^−1^) or the energy required (J.kg^−1^.min^−1^) and is fundamental to performance in long-distance running [57,58,59] because it shows the relative amount of sustainable capacity that is utilised at any given moment. For low work rates (e.g., below the first lactate turn point, LT1), the cost of exercise is relatively constant. At higher work rates, there is an additional and delayed cost that increases the energetic demands of locomotion [60,61,62].

Despite this clearly strenuous task, feedback from SG on the difficulty of the event was that she did not complain of real tiredness, but that the fatigue was akin to that she experienced in regular training. She expressed that her legs ached but felt that she recovered before each new marathon. This statement is similar to her experience during the previous world record where she set the record for running consecutively for 7 days [33]. SG stated that she did not find running a marathon every day for 10 days that challenging but suffered with boredom due to the monotony of the task, especially during the final 2 h of each marathon. It is well documented that pain tolerance is higher in ultra-runners [63], and as an experienced ultra-marathon runner SG is used to prolonged bouts of pain and discomfort, but strategies to alleviate boredom may have enhanced her performance further.

### 4.6. Strengths and Limitations

We were fortunate that SG’s record attempt took place in the same building as the Sports Science labs of Teesside University which allowed easy access to our equipment and simplified data collection. Many record attempts take place in remote locations, thus reducing data collection opportunities. This proximity led to the very detailed analysis of performance and changes in physiology that have rarely been reported in this detail for a live event where a world record of such long duration was broken.

Limitations to our work include estimations of EE throughout the day and using BIA for changes in body composition. BMR was calculated via BIA but did not include any EE measures for periods between the marathons, leading to a possible underestimation of total EE for the day. However, SG minimised her activity between attempts and only travelled to and from the venue (a short 15 min car journey). She rested for the remainder of the day and had her meals prepared for her. As such we are confident that total EE is not significantly different to that reported. We acknowledge that the use of BIA within athletic populations may be less reliable than in clinical populations when assessing change at the population level [64,65], however from a practical standpoint, given the high ICC of the reported device to dual-energy X-ray absorptiometry [42,43] we are satisfied that some degree of change that is meaningful to the athlete occurred.

## 5. Conclusions

We conclude that the successful record attempt was the result of a very stable pacing strategy, which was rigorously adhered to across all 10 marathons, carefully planned recovery strategies, and minimal physiological perturbations. The negligible fluctuation in the completion times highlight that SG was aware of her capabilities and confidently carried out her pre-determined race-pace, with only small variations in pace to alleviate monotony. SG favoured pre-race-, recovery-, and evening meals over in-race nutrition (due to nausea). Her excellent exercise economy and enhanced ability to metabolise fat as a fuel have previously been documented by us across seven days [33]. Although this clearly works for SG, it is not a strategy we would recommend to other athletes, as this would most likely lead to a performance drop after approximately three hours. Recommendations for similar events are, therefore, to plan and rigorously follow a realistic pace, ensure recovery strategies are in place immediately post-run, and allow sufficient periods for good quality sleep of long durations.

## Figures and Tables

**Figure 1 ijerph-18-12066-f001:**
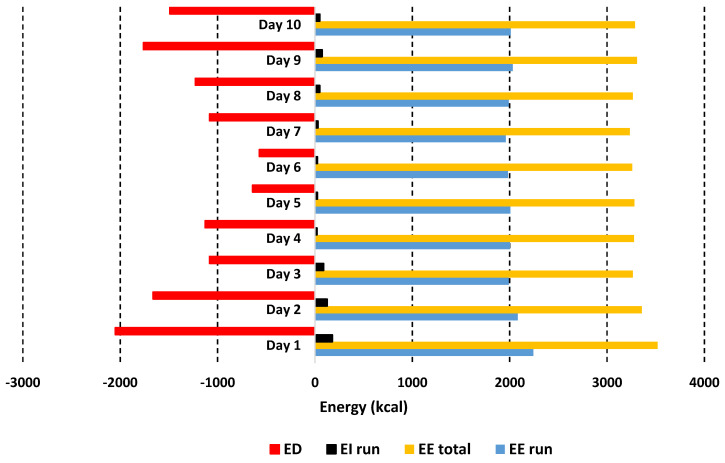
Energy intake (EI), energy expenditure (EE) and energy deficit (ED). Please note that the overall EI matches the EE during the runs, but overall, there is a significant ED every day due to EE from BMR and other activities.

**Figure 2 ijerph-18-12066-f002:**
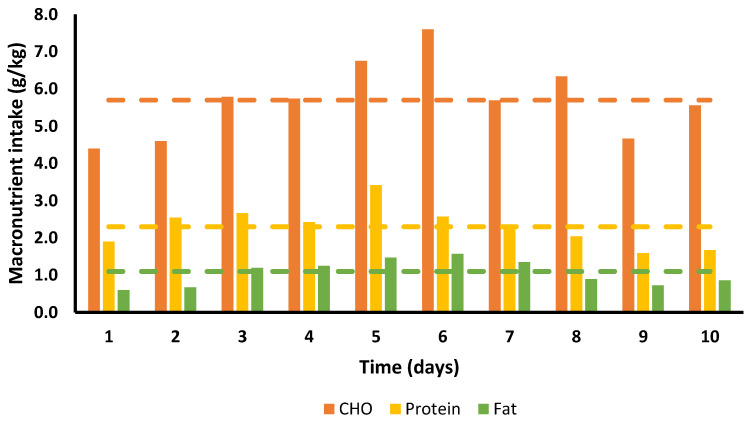
Daily macronutrient intake in grams per kg body mass. Bars represent individual macronutrient intake per day; dashed lines correspond to mean intake for each macronutrient.

**Figure 3 ijerph-18-12066-f003:**
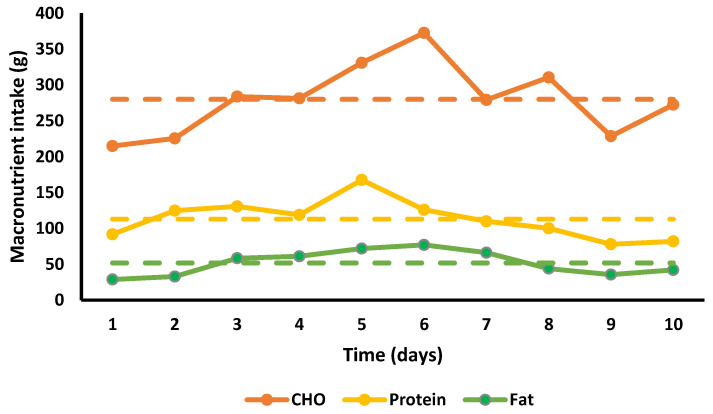
Total macronutrient intake in grams. Series lines link individual data points for the intake of each macronutrient (g) per day. Dashed lines represent the mean intake for each macronutrient across all 10 days.

**Figure 4 ijerph-18-12066-f004:**
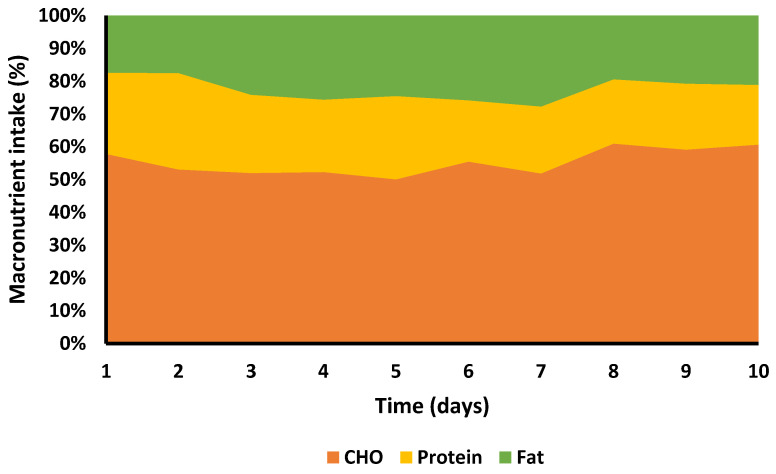
Percentage macronutrient contribution. Please note that mean carbohydrate contribution is 55% of total intake.

**Table 1 ijerph-18-12066-t001:** Completion times for all 10 Marathons.

Marathon	Completion Time
	(h, min, s)
1	4 h 21 min 21 s
2	4 h 21 min 39 s
3	4 h 24 min 38 s
4	4 h 24 min 06 s
5	4 h 23 min 55 s
6	4 h 24 min 22 s
7	4 h 23 min 54 s
8	4 h 23 min 21 s
9	4 h 23 min 25 s
10	4 h 21 min 21 s
Mean ^1^	4 h 23 min 9 s
Total	43 h 51 min 39 s

^1^ Please note the consistent finishing times, varying by just~3 min between attempts.

**Table 2 ijerph-18-12066-t002:** Heart rate for all 10 marathons.

Marathon	Heart Rate ± SD
	(b.min^−1^)
1	139 ± 10
2	140 ± 5
3	136 ± 9
4	146 ± 5
5	148 ± 10
6	142 ± 4
7	141 ± 7
8	142 ± 9
9	150 ± 8
10	144 ± 6
Mean ^2^	143 ± 4

^2^ Please note the highly consistent heart for all 10 marathons attempts. Only days 5 and 9 are slightly elevated compared to the mean.

**Table 3 ijerph-18-12066-t003:** Average heart rate per hour of marathon for all 10 marathons.

Hour of Marathon	Heart Rate ± SD
	(b.min^−1^)
1	137 ± 6
2	137 ± 5
3 ^3^	145 ± 6
4	151 ± 5
Mean	143 ± 4

^3^ Please note that despite the even pace there is a noticeable increase in heart rate from the 3rd hour.

**Table 4 ijerph-18-12066-t004:** Example food and drink for a typical day.

Breakfast	
Weetabix (oats)	2
Semi skimmed milk	100 mL
Sugar	2 teaspoons
Tea	200 mL
Coffee (instant)	200 mL
Actimel probiotic yoghurt	100 mL
10 a.m.	
Chicken cup soup	1
Bread, whole meal	1 slice
Event	
Grapes	16
Water	1750 mL
Post-marathon	
Semi skimmed milk	500 mL
Mountain fuel recovery powder	2 scoops
Neovite colostrum powder	1 teaspoon
Protein PHD pharma whey protein	35 g
Banana	1
Evening	
lamb, roast potatoes, parsnips, carrots, green beans, leaks, gravy	
apple crumble and double cream	
Tea	600 mL
Bed	
Milk	200 mL
Sugar	1 teaspoon
Instant drinking chocolate	1 teaspoon

## Data Availability

The data presented in this study are available on request from the corresponding author. The data are not publicly available due to the data in the present study pertaining to only one athlete.

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
