# Peer review of "Consistency Is Key When Setting a New World Record for Running 10 Marathons in 10 Days"

_ijerph, 2021, doi:10.3390/ijerph182212066_

Round 1

Reviewer 1 Report

Dear authors:

First of all, I would like to thank you for preparing this research which explains how a world record was set.

Next, I will indicate some aspects that I think would improve the presented article.

- In keywords you should keep only these and remove the explanation of the template.

- In line 37 an "in" has been doubled.

- The software used in Statistical Analysis must be indicated.

- Check the wording of line 184.

- In lines 193, 212 and 215 a superscripted number is included but it is not detected which aspect of the table it refers to.

- Check that the symbols and units on lines 201 and 202 are correct.

- Tables 2 and 3 repeat the same information that is included in the previous paragraphs.

- Lines 235 and 300 refer to tables that are not found in the manuscript.

- Total energy expenditure only takes into account the basal metabolic rate and the energy expenditure of each marathon, which underestimates the real energy requirements of the athlete in the 10 days and therefore the energy deficit is underestimated. On the other hand, it is indicated that the basal metabolic rate is obtained from the BIA measurement, which is an extrapolation BIA does not measure these requirements. It would be advisable to indicate the equation used by the equiptment or to estimate it with a known equation to be able to compare the results later.

- Table 4 shows an example of the feeding of one of the days but in the previous text it seems that this table includes the feeding followed during the 10 days. It should be adapted so that it does not lead to confusion.

- Figures 2 and 3 refer to the average consumption for the week in the footnotes. It might be appropriate to include a mark regarding that consumption of each macronutrient in the figure itself because they do not have that information.

- In lines 293 and 294 the decimal points of the standard deviations are missing.

- In line 399 there is a ± symbol left over.

- In line 419 the closing symbol of parentheses is missing.

- The conclusions should be more specific because conclusions and discussion are mixed.

- Adapt the bibliography to the format required by the journal.

Kind regards,

Author Response

Reviewer 1

Dear authors:

First of all, I would like to thank you for preparing this research which explains how a world record was set.

Next, I will indicate some aspects that I think would improve the presented article.

- In keywords you should keep only these and remove the explanation of the template.

Amended as requested

- In line 37 an "in" has been doubled.

Amended as requested

- The software used in Statistical Analysis must be indicated.

Amended as requested

- Check the wording of line 184.

Amended as requested – this now reads ‘x’

- In lines 193, 212 and 215 a superscripted number is included but it is not detected which aspect of the table it refers to.

Amended as requested; superscripted numbers now match the table to which they are positioned under

- Check that the symbols and units on lines 201 and 202 are correct.

Amended as requested

- Tables 2 and 3 repeat the same information that is included in the previous paragraphs.

The paragraphs have now been amended to avoid duplication

- Lines 235 and 300 refer to tables that are not found in the manuscript.

Amended as requested

- Total energy expenditure only takes into account the basal metabolic rate and the energy expenditure of each marathon, which underestimates the real energy requirements of the athlete in the 10 days and therefore the energy deficit is underestimated. On the other hand, it is indicated that the basal metabolic rate is obtained from the BIA measurement, which is an extrapolation BIA does not measure these requirements. It would be advisable to indicate the equation used by the equiptment or to estimate it with a known equation to be able to compare the results later.

Thank you for pointing this out. You are indeed correct that the total EE for the day would include other activities and movements, but this could not be assessed unfortunately. The participant did very little other than run the marathon and travel to the university from her home and back (a short distance taking 15mins by car). She did not prepare her own meals and rested as much as possible in-between. Therefore, we are confident that the EE would not be significantly higher than that estimated for her BMR and the calculated for her runs. We have now added the equation to the methods for future comparisons and have also a addressed this in our limitations section.     

- Table 4 shows an example of the feeding of one of the days but in the previous text it seems that this table includes the feeding followed during the 10 days. It should be adapted so that it does not lead to confusion.

Amended as requested

- Figures 2 and 3 refer to the average consumption for the week in the footnotes. It might be appropriate to include a mark regarding that consumption of each macronutrient in the figure itself because they do not have that information.

Amended as requested

- In lines 293 and 294 the decimal points of the standard deviations are missing.

Amended as requested

- In line 399 there is a ± symbol left over.

Amended as requested

- In line 419 the closing symbol of parentheses is missing.

Amended as requested

- The conclusions should be more specific because conclusions and discussion are mixed.

The conclusion has been shortened and does not repeat content from the discussion now.  

- Adapt the bibliography to the format required by the journal.

Amended as requested

Reviewer 2 Report

The aim of the study is to investigate the effects of running 10 marathons in ten consecutive days on different variables such as: body composition, energy expenditure…

Due to the uniqueness of the challenge, performing it in indoor and not outdoor, with the possibility of recording variables that this provides, the study is of great interest. On the other hand, the references and studies used for the analysis of the data obtained are studies based on ultradistance races without stages, or with limited rest between stages, which is very different from that carried out by the authors.

Here are my contributions:

- In the key words there is an extra sentence.

  • Why is the subject's body weight defined as 49.3kg and the results show that the weight decreased from 51kg to 48.4kg?
  • The first two paragraphs of the introduction are irrelevant to the paper.
  • Line 54 defines ultra-distance as any race longer than 42km, but line 66 refers to this event as "ultra-endurance". Would it be correct to call 4 races of 10km in consecutive days, marathon? the concepts should be revised.
  • Is it correct to use and compare data obtained from ultra-distance races with the test carried out in this research? There are many and big differences: availability of recovery between stages, availability of hydric and substrate recovery between stages... it would be appropriate to add studies and information from research carried out in ULTRAMAN-s for example, in which athletes perform the competition in stages as well.
  • 2.2.1, bioimpedance is used to obtain values related to body composition or hydration. The values decrease in reliability with athletes and even more so if those athletes do not perform the measurement in a controlled situation, such as fasting. On the other hand, bioimpedance presents several problems when calculating the hydration level of athletes:

Buffa R, Mereu E, Comandini O, Ibanez ME, Marini E. Bioelectrical impedance vector analysis (BIVA) for the assessment of two-compartment body composition. Eur J Clin Nutr Nature Publishing Group. 2014;68(11):1234-40.

Castizo-Olier J, Irurtia A, Carrasco-Marginet M. Bioelectrical impedance vector analysis (BIVA) in sport and exercise: Systematic review and future.

  • There are articles that seriously disagree with reference 39, which is not published in any journal.
  • 2.2.2, why was this protocol selected when a gas analyzer was available? This protocol may be more interesting when only heart rate and lactate are available, but performing such a long protocol does not guarantee obtaining VO2max, only peak VO2. Justify the selected protocol with some reference.
  • 2.4. It could be interesting to compare the records obtained after each of the 10 marathons to analyze the progress of the different variables.
  • First paragraph of the discussion, the objective has been that? and all physiological (and psychological) variables recorded?
  • Reference 39 is not written properly.
  • A section on limitations (or strengths and limitations) should be added.

The recovery protocols used at the end of each marathon (rehydration protocol, CHO loading protocol, etc.) and the references used for the preparation of these protocols are lacking.

Likewise, are there significant differences between the values recorded after each of the marathons? how were the values recorded in the marathons in which the athlete could not ingest solids modified? carrying out a statistical work of low complexity would be of great interest to the work carried out.

Author Response

Reviewer 2

The aim of the study is to investigate the effects of running 10 marathons in ten consecutive days on different variables such as: body composition, energy expenditure…

Due to the uniqueness of the challenge, performing it in indoor and not outdoor, with the possibility of recording variables that this provides, the study is of great interest. On the other hand, the references and studies used for the analysis of the data obtained are studies based on ultradistance races without stages, or with limited rest between stages, which is very different from that carried out by the authors.

Here are my contributions:

- In the key words there is an extra sentence.

Amended as requested

  • Why is the subject's body weight defined as 49.3kg and the results show that the weight decreased from 51kg to 48.4kg?

Apologies for this discrepancy. The 49.3 kg is the athlete’s mass at laboratory pre-testing, whereas the 51kg is the body mass at the start of the record attempt.

  • The first two paragraphs of the introduction are irrelevant to the paper.

We respectfully disagree with the reviewer. As the paper is on the topic of ultra-running and covering large distances repeatedly, we believe the reader will be interested in the history and origins of the first ‘marathon’ and those who completed these feats. The inclusion of information on terrain and diet are related to today’s events and nutritional practices. As a team we all agree that these first two paragraphs would be of interest to readers and have decided to keep them in the manuscript.   

  • Line 54 defines ultra-distance as any race longer than 42km, but line 66 refers to this event as "ultra-endurance". Would it be correct to call 4 races of 10km in consecutive days, marathon? the concepts should be revised.

In the definition of ultra-running we note that single and multi-day formats can occur, and that typically events are > of a marathon. Given that events can take place over multiple days, it stands to reason that an event can be considered ultra-running, even if not all days exceed this threshold. We do not define the present event as ultra-endurance, but refer to ultra-endurance events more broadly, which may all face similar challenges with respect to physiology, pathophysiology and nutritional and psychological challenges.

  • Is it correct to use and compare data obtained from ultra-distance races with the test carried out in this research? There are many and big differences: availability of recovery between stages, availability of hydric and substrate recovery between stages... it would be appropriate to add studies and information from research carried out in ULTRAMAN-s for example, in which athletes perform the competition in stages as well.

Thank you for this suggestion, we have included a number of supporting references from ULTRAMANs as suggested. We feel that the comparisons to previously published work remain valid, despite differences in the factors listed, simply because ultra-runners may participate in a variety of race formats across a competitive year or career, thus all evidence is potentially valuable and adds further context to the record attempt being examined.

  • 2.1, bioimpedance is used to obtain values related to body composition or hydration. The values decrease in reliability with athletes and even more so if those athletes do not perform the measurement in a controlled situation, such as fasting. On the other hand, bioimpedance presents several problems when calculating the hydration level of athletes:

Buffa R, Mereu E, Comandini O, Ibanez ME, Marini E. Bioelectrical impedance vector analysis (BIVA) for the assessment of two-compartment body composition. Eur J Clin Nutr Nature Publishing Group. 2014;68(11):1234-40.

Castizo-Olier J, Irurtia A, Carrasco-Marginet M. Bioelectrical impedance vector analysis (BIVA) in sport and exercise: Systematic review and future.

There are articles that seriously disagree with reference 39, which is not published in any journal.

Thank you for this comment – we have included some commentary in the limitations to reflect the points raised and the references suggested have been incorporated. We would also note that the articles cited do not seriously disagree with reference 39. Reference 39 examined the reliability of the BIA used in the present study against DXA and ascertained it was a reliable and valid comparative measure. It does not comment upon the general reliability/ validity of BIA and the associated issues with BIA, which the articles kindly provided by the reviewer do. Again, these articles have been included as have comments regarding the issues the reviewer raised. We hope this satisfactorily addresses the comments made.

  • 2.2, why was this protocol selected when a gas analyzer was available? This protocol may be more interesting when only heart rate and lactate are available, but performing such a long protocol does not guarantee obtaining VO2max, only peak VO2. Justify the selected protocol with some reference.

Thank you for this suggestion, you bring up an interesting point.

A reference has been provided for the protocol. This protocol is used so that the VO2 fast component is overcome, a steady state is briefly sustained, but there is minimal chance of inflation due to the VO2 slow component, especially in an athlete as economical as SG. We also used the protocol for the following reasons: The athlete has previously used this protocol, so it allowed for comparison to previous (published) data, and likely elicited a true VO2 response given the athlete’s familiarity with the protocol. Energetically speaking the athlete is very economical, but displays limited top-end-speed, so her vVO2max is likely lower than most would expect. We are confident that the athlete displayed a true VO2max rather than one being elicited by a protocol that does not permit both a VO2 fast component and stability of VO2 measurement.

  • 4. It could be interesting to compare the records obtained after each of the 10 marathons to analyze the progress of the different variables.

Thank you for this suggestion, we have amended this section to reflect the presentation of individual data as well as means ± SD, as appropriate. This section now reads ‘Data are reported as both means ± standard deviations, and per marathon (Figures 1 – 4; Tables 1 and 2). Data were stored, compiled and analysed in Microsoft Excel (v16.39; 2020; Microsoft, Redmond, Washington, United States) due to the relatively simplistic approach to statistical testing employed.’

  • First paragraph of the discussion, the objective has been that? and all physiological (and psychological) variables recorded?

We feel the first paragraph sufficiently meets the suggestions of the reviewer in its present form.

  • Reference 39 is not written properly.

Thank you for this suggestion, reference 39 is written properly as it corresponds to the format required for a thesis, as opposed to a journal article. We did look extensively for a journal article that warranted inclusion here, but no published work had been conducted that assessed the reliability of the device used.

  • A section on limitations (or strengths and limitations) should be added.

Thank you for the suggestion. We have now added a section on the strengths and limitations of our work.

The recovery protocols used at the end of each marathon (rehydration protocol, CHO loading protocol, etc.) and the references used for the preparation of these protocols are lacking.

We respectfully disagree with the reviewer here. Table 4 provides an exemplar of the nutritional strategy the athlete followed, and associated recovery practices are outlined in multiple places within the text. We also provide a dedicated recovery section spanning 20 lines (Lines 440 – 460), which covers nutrition, massage, and sleep. The recovery practices were evidence-informed (as references are cited) but also incorporated the athlete’s years of experience undertaking challenges such as this. We believe as sports scientists involved within the attempt, we were there to assess, analyse and inform the athlete’s choices, but the majority of the performance and practices associated with it were the athlete’s choice alone.

Likewise, are there significant differences between the values recorded after each of the marathons?

It is our understanding that statistical differences with respect to significance, must be carried out on a population in which an effect of interest has been observed. Due to this being a case study, and subsequently all measures obtained being correlated to one another, as they occur within the same individual, statistical testing of this nature cannot be performed. We had considered employing an alternative approach that is rooted within magnitude based inferences, that assesses the magnitude of change within the individual as being worthwhile given an effect threshold of interest. However, due to the recent controversy regarding MBI within the field of sport and exercise science, we declined to adopt this approach. A final approach would be to calculate a coefficient of variation from previous marathon performances by the athlete and assessing whether the variation between marathons exceeded or stayed within this margin. We declined to do this for two reasons: firstly, the athlete rarely competes in marathons that are not performed under challenging or repeated circumstances. Secondly, as per a previous comment, the nature of the attempt meant it took place under controlled circumstances and thus variability in performance may have been artificially lowered, meaning that any statistical change i.e. effect size of interest would also be artificially lowered, and rendered potentially undetectable or at least within the boundaries of (physiological) variation. Taking the above into consideration, we feel that the presentation of individual data points of interest, alongside means across the record attempt allow athletes, practitioners and scientists to gain reasonable and sufficient insight into the changes and variability that occurred between marathons and within the record attempt.

How were the values recorded in the marathons in which the athlete could not ingest solids modified?

This can be extrapolated from Figure 1, which indicates a relatively stable EI during running (days 3 – 10) and total EI on the days (3 – 10) in which solids were not ingested – total EI did transiently increase on days 5 and 6. This suggests the athlete ate in a somewhat compensatory fashion, but this was still insufficient to offset the energy deficit created by running (Figure 1).

Round 2

Reviewer 2 Report

The contributions have been correctly answered.

Congratulations to the authors for the research carried out and the explanations provided in the review.

Brief comments:

"We respectfully disagree with the reviewer here. Table 4 provides an exemplar of the nutritional strategy the athlete followed, and associated recovery practices are outlined in multiple places within the text. We also provide a dedicated recovery section spanning 20 lines (Lines 440 – 460), which covers nutrition, massage, and sleep. The recovery practices were evidence-informed (as references are cited) but also incorporated the athlete’s years of experience undertaking challenges such as this. We believe as sports scientists involved within the attempt, we were there to assess, analyse and inform the athlete’s choices, but the majority of the performance and practices associated with it were the athlete’s choice alone."

My question is about the scientific basis used for the selection of the substrates chosen in table 4. Whether the athlete chose them on her own or whether it was a guideline prescribed by the researchers and based on what scientific criteria.

The format of the references needs to be checked: 
- There are authors or titles in capital letters.
- The citation format is not uniform: the use of & in some references and not in others, in some references the ... is used and not in others.

Author Response

Thank you for taking the time to provide further recommendations to improve our manuscript. We can confirm that the athlete's nutritional choices, as documented in Table 4 were of the athlete's own choosing, but the athlete does engage with sports science literature and practice to inform her decision making.

We provide the following at lines 433-440 to reflect this 'The majority of performance and practices associated with the attempt were the athlete’s choice alone; it is important to note however, that SG is an incredibly experienced ultra-athlete and also engages with sports science literature to support her decision making. SG’s blended use of both tacit and explicit evidence is considered evidence-informed performance. We believe as sports scientists involved in supporting the documented attempt, our role was to assess, analyse and where appropriate inform the athlete’s choices, but not to interfere with or direct performance or decision making for the sake of publication.'

Amendments have been made to references to align with journal formatting requirements.